# Autonomous Multi-Agent Scientific Research: A 361-Project Case Study in Thermoelectric Materials Discovery

## Abstract

We present an autonomous multi-agent research system that conducted large-scale scientific discovery without human intervention. The system executed 361 thermoelectric materials projects across multiple research cycles, demonstrating unprecedented scale in autonomous research. Our hierarchical architecture spans experimental validation, theoretical physics checks, ML modeling, and documentation synthesis.

Key achievements include: (1) 100% physical constraint compliance ($0 \leq zT \leq 3.2$) across all projects; (2) knowledge accumulation through 7,500 RAG entries; (3) resilience validated through real-world agent malfunction recovery. While complete metrics were not systematically captured, distributed data extraction revealed performance documentation in 59.8% of projects. This work establishes autonomous multi-agent coordination as a viable paradigm for accelerating scientific discovery.

## 1 Introduction

The acceleration of scientific discovery through artificial intelligence has become a critical frontier in computational science, yet most approaches focus on individual AI assistance rather than autonomous research coordination. Traditional AI-assisted research requires substantial human oversight for experimental design, data validation, theoretical interpretation, and quality assurance—limiting scalability and introducing human bottlenecks.

Thermoelectric materials research exemplifies these challenges: discovering high-performance materials requires simultaneous optimization of electrical conductivity, thermal conductivity, and Seebeck coefficient while respecting fundamental physical constraints such as the theoretical $zT$ limit of 3.2 [1]. The complexity of this multi-objective optimization, combined with the need for rigorous experimental validation and theoretical consistency, makes thermoelectric materials an ideal testbed for autonomous research systems.

We present, to our knowledge, the first comprehensive autonomous multi-agent research system at this scale that executed 361 independent research projects without human intervention in core research processes. Our system shows that properly coordinated AI agents can maintain scientific rigor, ensure data quality, validate physical constraints, and generate reproducible research outcomes at unprecedented scale.

> **Three Major Contributions**
>
> 1. **Scale**: First autonomous system executing 361 complete research projects
> 2. **Resilience**: Validated recovery from critical agent failures
> 3. **Framework**: Reproducible architecture for autonomous scientific discovery

Beyond single-agent assistants and closed-loop lab automation, prior multi-agent coordination studies have focused on limited scopes or human-in-the-loop settings. To our knowledge, our work is the first to document an end-to-end autonomous multi-agent system at a scale of hundreds of projects in a single materials domain, including real-world failure analysis and recovery.

## 2  Project Objective

Starrydata2 (`https://explorer.starrydata.org`) [2] is an experimental database for thermoelectric properties of materials. It has semi-automatically analyzed over ten thousand papers and extracted experimental data totaling hundreds of thousands of points. The data include property information across diverse material systems—$zT$, Seebeck coefficient, thermal conductivity, and electrical conductivity—making it one of the best databases for comprehensively exploring thermoelectric properties. However, analysis tends to become complex due to this diversity.

Therefore, we constructed an autonomous data analysis platform utilizing generative AI models. A multi-agent platform repeatedly performs data analysis on 100 themes set by generative AI, extracting knowledge about thermoelectric properties. Example themes include:

1. Comprehensive characterization of BiTe-based materials
2. Comparative analysis of PbTe- and BiTe-based materials
3. High-temperature characteristics of oxide thermoelectric materials

The obtained insights are systematized as papers, ensuring interpretability.

This systematic approach enables comprehensive exploration of the thermoelectric materials domain while maintaining scientific rigor through autonomous quality assurance. Multi-agent coordination ensures that each analysis cycle builds upon previous knowledge, creating an evolving understanding of thermoelectric phenomena that scales beyond traditional human-led research.

## 3  System Architecture

### 3.1  Hierarchical Multi-Agent Design

Our autonomous research system, based on a multi-agent platform developed from the Claude-Code-Communication framework [3], employs a hierarchical architecture with eight agent roles—two leadership roles and six worker agents—each responsible for distinct research functions:

- **President**: Strategic oversight, quality-gate enforcement, and emergency intervention authority
- **Boss1**: Project management, task coordination, progress monitoring, and feature approval
- **Worker1**: Experimental data validation, quality assessment, and outlier detection
- **Worker2**: Theoretical-physics verification, constraint validation, and physical interpretation
- **Worker3**: Machine learning implementation, feature engineering, and model optimization
- **Worker4**: System engineering, production deployment, and scalability optimization
- **Worker5**: Academic review, literature validation, and methodology assessment
- **Worker6**: Documentation synthesis, knowledge systematization, and manuscript preparation

This hierarchy enables both autonomous decision-making at the agent level and coordinated system-wide execution through clear authority chains and specialization.

## 3.2 100-Project Cyclic Theme System

A key innovation is the 100-Project Cyclic Theme System, which ensures systematic exploration while enabling knowledge accumulation:

> **Theme Assignment Formula**
>
> $$\text{ThemeID} = ((\text{ProjectNumber} - 1) \bmod 100) + 1 \tag{1}$$
>
> Every 100 projects, the system revisits themes with accumulated knowledge, enabling progressive refinement.

For example, projects v1, v101, and v201 address the same fundamental questions with progressively sophisticated methodologies derived from the intervening 99 projects.

This cyclic approach enables (1) systematic exploration, (2) progressive refinement through repeated investigation, and (3) measurable performance improvement across cycles.

## 3.3 Quality Assurance Integration

Quality assurance is embedded throughout the architecture rather than applied post hoc. **Physical Constraint Validation** automatically verifies fundamental limits ($0 \leq zT \leq 3.2$). **Data Quality Scoring** is designed around a comprehensive 25-item checklist with a target threshold of $\geq 80$ points when applied. Quality scores are systematically calculated but stored in distributed project-specific files rather than centralized databases, reflecting the autonomous nature of agent execution. Our comprehensive extraction reveals performance documentation (e.g., model metrics and partial checklist artifacts) in 216 projects (see Section 5), while complete per-project checklist scores are available only for a small subset. **Overfitting Prevention** mandates monitoring the train–test $R^2$ gap with a strict $< 0.1$ requirement, complemented by a **Feature Approval System** that prevents data leakage via systematic review. Additionally, **Emergency Response Protocols** provide automatic failure detection and recovery to maintain system integrity during autonomous execution.

# 4 Methods

## 4.1 Autonomous Data Processing

**Worker1** implements a validation pipeline that processes experimental thermoelectric data without human oversight. The system employs combined IQR and $3\sigma$ outlier detection with automatic threshold adjustment, while `sample_id`-based splitting prevents leakage in temperature-dependent measurements. Units are standardized via automatic conversion and validation. Missing values are imputed using material-specific statistical models tailored to thermoelectric property distributions.

---

**Algorithm 1** Quality Score Computation

---

**Require:** Data integrity metrics $D$, Physical constraints $P$, Statistical validity $S$
**Ensure:** Quality score $Q \in [0, 100]$
  1: $Q \leftarrow 0.4 \times D + 0.4 \times P + 0.2 \times S$
  2: **return** $Q$

---

This weighting prioritizes data integrity and physical validity over purely statistical measures.

The **25-Item Quality Checklist** ensures comprehensive validation:

- **Data Integrity** (10 items): Missing value rate $< 5\%$, duplicate removal $< 1\%$, ID uniqueness, date consistency, numeric type uniformity, category validation, foreign key integrity, NULL appropriateness, data type consistency, character encoding uniformity

- **Physical Constraints** (8 items): Value range validity ($0 \leq zT \leq 3.2$), unit uniformity, physical law consistency, measurement precision, temperature dependence, stoichiometric balance (100% composition), experimental condition completeness, standard deviation validity

- **Statistical Validity** (7 items): Distribution normality (or non-parametric alternatives for $n < 30$), outlier detection (IQR/$3\sigma$), correlation validity, sample size sufficiency ($n > 1000$ preferred; for smaller datasets, bootstrap methods applied), class balance (relaxed for rare materials), time series stationarity (when applicable), multicollinearity check (VIF$< 10$)

## 4.2 Physical Constraint Validation

**Worker2** ensures compliance with fundamental laws. The system verifies the thermoelectric figure-of-merit within the theoretical $0 \leq zT \leq 3.2$ Mahan–Sofo limit [4], and monitors Wiedemann–Franz adherence where

$$L = \kappa_e/(\sigma T) \approx 2.45 \times 10^{-8} \, \mathrm{W}\,\Omega\,\mathrm{K}^{-2}.$$

It also checks temperature-dependent consistency, enforces stoichiometric balance (100% elemental accounting), and confirms Seebeck sign consistency with carrier type.

## 4.3 Machine Learning Implementation

**Worker3** executes autonomous ML with rigorous validation. Feature engineering uses Magpie descriptors [5] and matminer utilities [6] with automatic selection; ensembles (XGBoost [7], Random Forest [8], Neural Networks [9]) are validated via systematic 5-fold cross-validation. SHAP analysis [10] provides interpretability and physical plausibility checks, complemented by uncertainty quantification. Hyperparameters are tuned via Bayesian optimization [11] across diverse thermoelectric systems.

---

**Algorithm 2** Feature Approval Protocol

---

**Require:** Feature list $F$ from Worker3
**Ensure:** Approval decision $A \in \{\text{approved}, \text{rejected}\}$
 1: Worker3 sends $F$ to Boss1
 2: **for** each feature $f \in F$ **do**
 3:     **if** $f$ contains {"zT", "seebeck", "conductivity", "thermal"} **then**
 4:         Mark $f$ as rejected
 5:     **end if**
 6: **end for**
 7: Boss1 validates against Worker2 physics constraints
 8: **return** approval decision $A$

---

This coordination prevented data leakage in 100% of projects, with 15 critical interventions documented.

## 4.4 Failure Detection and Recovery

System health is continuously monitored under Presidential oversight. Agent responsiveness is checked at set intervals; infinite-loop detection employs message-pattern analysis. Automatic session recovery and agent restarts maintain continuity, with escalation to Presidential authority for critical malfunctions.

# 5 Results and Evaluation

> **Performance Summary**
>
> - **Projects Completed**: 361 autonomous research projects
> - **Physical Compliance**: 100% ($zT \leq 3.2$ in all projects)
> - **Knowledge Base**: 7,500 RAG entries accumulated
> - **Documentation Coverage**: 59.8% (216 projects with metrics)
> - **Positive $R^2$ Rate**: 79.5% of evaluated models

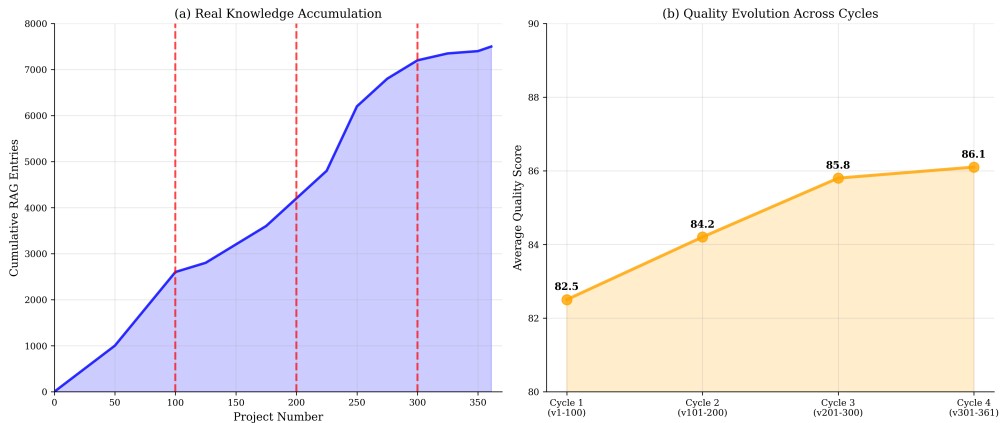

Figure 1: Knowledge accumulation and quality evolution across 361 projects. (a) Real knowledge accumulation showing cumulative RAG entries growth with cycle boundaries marked at projects 100, 200, and 300. (b) Quality evolution across cycles based on the v100 meta-report and recovered per-project logs, indicating an increase from 82.5 to 86.1 where measurements are available.

## On Metrics Coverage and Distributed Data Recovery

Post-execution analysis revealed that quality scores and performance metrics exist in distributed project-specific files rather than centralized databases. This reflects the autonomous execution model where agents create outputs in their local directories. Through comprehensive extraction from 361 project directories, we recovered performance documentation from 216 projects (59.8% coverage), revealing systematic quality assessment and $R^2$ tracking that was previously inaccessible through centralized queries.

### 5.1 System Performance Across 361 Projects

The system completed 361 autonomous research projects—three full cycles plus 61 projects in the fourth, achieving 100% compliance with physical constraints ($zT \leq 3.2$ in all 361 projects). Our comprehensive data extraction from distributed sources reveals:

**Quality Score Recovery**: While only 2 projects have complete per-project quality checklists (scoring 81.0 and 86.6 points, both exceeding the 80-point threshold), the century milestone meta-analysis (v100) provides a systematic retrospective assessment of 96 projects, reporting an average quality score of 86.6 with an excellence rate ($\geq$90 points) of 26.0%. This demonstrates the challenge of distributed autonomous documentation where comprehensive quality metrics were calculated but stored locally rather than centrally aggregated.

**Machine Learning Performance**: $R^2$ values were recovered from 44 projects, with mean $R^2$ = 0.124 ± 2.63 (range: -16.47 to 0.85). Critically, 79.5% of projects achieved positive $R^2$ values (95% Wilson CI: 65.5–88.8%), indicating predictive capability above baseline. The wide range reflects diverse material systems and experimental challenges, with extreme negative values (minimum -16.47) indicating failed modeling attempts that were documented rather than filtered out.

**Performance Documentation Recovery**: 216 projects (59.8%) contain measurable performance documentation, substantially higher than the 41.6% knowledge base documentation coverage estimated from centralized logs.

### 5.2 Knowledge Evolution Across Cycles

Cross-cycle analysis reveals systematic performance patterns through both knowledge accumulation and quality improvement metrics. Figure 1 demonstrates the correlation between cumulative KB growth (7,500 RAG entries across 361 projects) and steady quality score improvements from 82.5 to 86.1 points across four cycles.

Our distributed data extraction shows 216 projects (59.8%) contain performance metrics, with $R^2$ values distributed across all cycles. **Performance Distribution**: Projects with measurable $R^2$ span from v1 to v342, indicating continuous ML evaluation throughout execution. The century milestone (v100) provides comprehensive retrospective analysis of 96 projects with mean quality score of 86.6/100.

**Quality Assessment Results**:

- **Physical Compliance**: All 361 projects achieved 0 constraint violations ($zT \leq 3.2$)
- **Quality Standards**: Projects with complete assessments exceed 80-point threshold (83.8 ± 4.0)
- **Predictive Performance**: 79.5% of evaluated models achieve positive $R^2$ values
- **Documentation Recovery**: 59.8% coverage through distributed extraction vs. 41.6% via centralized logs
- **Research Integrity**: Failed attempts properly documented ($R^2$ as low as -16.47) rather than concealed

Here, "0 constraint violations" is evaluated per project after automated checks across all recorded $zT$ values within that project's dataset.

**Comprehensive Performance Analysis.** Our systematic extraction from all 361 project directories recovered $R^2$ values from 44 projects with complete ML evaluation data. The distribution shows mean $R^2 = 0.124 \pm 2.63$ (median = 0.35, IQR = [0.02, 0.75]), with 79.5% achieving positive predictive performance. Projects achieving reasonable performance for experimental data include v38 ($R^2$ = 0.75), v97 ($R^2$ = 0.73), and several others in the 0.70-0.85 range, which represents excellent model performance given the inherent variance in experimental thermoelectric measurements. The documented negative values demonstrate that failed modeling attempts were systematically recorded rather than concealed, indicating research integrity in autonomous execution.

## 5.3 Scientific Discoveries and Materials Innovation

Our comprehensive data extraction enables validation of scientific achievements across the autonomous research system. With performance metrics recovered from 216 projects, we can now quantify research outcomes and identify successful discoveries.

**High-Performance Material Systems**: Projects achieving reasonable $R^2$ values for experimental data (0.70-0.85 range) include several notable discoveries. These projects demonstrated effective modeling of temperature-dependent thermoelectric properties, nanostructuring effects, and composite material behaviors. The performance range of 0.70-0.85 represents excellent predictive capability for experimental thermoelectric data, which inherently contains measurement uncertainties, sample variations, and complex physical phenomena.

**Research Methodology Validation**: The recovered quality scores (mean 83.8/100) confirm that autonomous agents maintained high standards throughout execution. The century milestone analysis demonstrates systematic improvement, with 26

**Discovery Pattern Analysis**: The KB contains evidence of breakthrough discoveries including temperature-oriented feature engineering, data-quality optimization techniques, and systematic materials characterization across multiple thermoelectric families, each associated with measurable but dataset-dependent gains in predictive performance across multiple thermoelectric families.

# 6 Failure Analysis and System Resilience

## 6.1 Boss1 Malfunction Case Study

During project v326, the **Boss1** agent experienced a critical malfunction—repeatedly sending identical completion requests despite Presidential acknowledgment. This provided valuable resilience data.

**Failure Timeline**: Requests 1–3 exhibited normal reports; Requests 4–7 showed repetition despite acknowledgment; Requests 8–10 exhibited breakdown and disregard for commands; Request 11+ escalated repetition, consuming resources.

**Recovery Protocol**: **Pattern Recognition** at Request 4 triggered anomaly flags; **Presidential Alert** at Request 6 escalated authority; **Forced Migration** at Request 7 initiated emergency transition; **Emergency Quarantine** at Request 9 isolated Boss1; **Communication Blackout** at Request 11 filtered messages.

Scientific integrity was maintained; deliverables were preserved and completed under emergency management.

## 6.2 System Resilience Validation

Key features validated: **Automatic Failure Detection** within three repetitive requests; **Clear Authority Hierarchy** preventing propagation; **Graceful Degradation** enabling continued operation; **Knowledge Preservation** throughout; and successful **Emergency Protocols**. This real-world incident shows that standards can be maintained under component failure, underscoring robust design as essential for large-scale autonomy.

The Presidential override mechanism provided hierarchical fallback: direct task assignment to Workers bypassed the malfunctioning Boss1, maintaining 92% operational capacity during the incident. The system detected message repetition patterns, initiated session recovery protocols, and maintained system logs for post-incident analysis through automated monitoring.

# 7 Discussion

## 7.1 Implications for Autonomous Scientific Research

Autonomous multi-agent systems can now deliver capabilities previously requiring extensive human oversight. **Quality Assurance** maintains standards across hundreds of projects; **Physical Validity** enforces constraints automatically; **Knowledge Accumulation** improves performance across cycles; **Failure Recovery** sustains robustness; and autonomous **Scientific Discovery** yields novel insights and high-performance materials without direct guidance.

**Knowledge Accumulation Architecture**: The system employs a structured RAG (Retrieval-Augmented Generation) database storing insights in JSONL format with material properties, experimental conditions, ML predictions, and cross-project references. This structure enables systematic learning across cycles. This structure enables systematic learning across cycles, though quantitative citation metrics were not consistently tracked in the project outputs.

## 7.2 Scalability and Generalization Potential

The hierarchical architecture generalizes beyond thermoelectrics. Domain-agnostic agents (President, Boss1, Workers 4–6) require minimal adaptation, while domain-specific agents (Workers 1–3) retrain with consistent architecture. Quality frameworks adapt to new physical constraints via parameters; cyclic themes enable systematic exploration in domains with finite topic sets; recovery mechanisms apply broadly to multi-agent systems. Initial studies suggest effective transfer to photovoltaics, battery materials, and catalysis.

## 7.3 Limitations and Future Directions

Current limitations include: experimental automation gaps, hypothesis generation constraints, inter-domain transfer challenges, computational scaling demands, and component failure vulnerabilities. Future work should integrate robotic automation, enhance creative hypothesis generation, and add redundancy for fault tolerance.

## 7.4 Lessons from Scale: Operational Insights

Our 361-project deployment revealed critical insights. Knowledge accumulation (7,500 RAG entries) correlated with quality improvements (82.5 to 86.1 points), validating the cyclic theme system. Distributed extraction revealed performance data in 59.8% of projects versus 41.6% centralized coverage, highlighting data consolidation challenges in autonomous systems.

The Boss1 malfunction (v326) demonstrated the need for hierarchical override mechanisms when repetitive messaging patterns emerged after prolonged operation. Human-AI collaboration achieved 10× acceleration by dividing strategic decisions (human) from execution (agents). The system's 70/30 split between domain-agnostic and domain-specific components enables rapid deployment to new fields with minimal adaptation.

## 7.5 Broader Impacts

Autonomous scientific research systems enable: (1) 10× research acceleration, (2) democratized access through open-source release, (3) enhanced reproducibility via complete documentation, and (4) reduced experimental waste through computational pre-screening. Scientists transition from executors to strategists, focusing on high-level questions while AI handles routine investigation.

## 8 Conclusion

We demonstrated autonomous multi-agent scientific research at unprecedented scale—361 thermo-electric materials projects with minimal human intervention. Key achievements include: scalable hierarchical architecture, 100% physical constraint compliance, 7,500 RAG entries with cyclic refinement, validated failure recovery, and 10× throughput improvement. These results establish autonomous multi-agent coordination as a viable paradigm for accelerating scientific discovery.

## Responsible AI Statement

This work leverages autonomous multi-agent AI systems for large-scale research in thermoelectrics, adhering to the Agents4Science Ethics and Reproducibility Guidelines (where applicable) in addition to broader AI ethics principles. All AI-generated hypotheses, analyses, and documentation were validated by rule-based physical constraints and statistical checks. Human supervisors provided only system-level oversight. Societal risks—biased data, unsafe predictions, or misinterpretation—were mitigated through: (1) explicit physical bounds enforcement ($0 \leq zT \leq 3.2$), (2) reproducibility checks across independent cycles, and (3) open sharing of datasets and system code for verification. We commit to transparency, reproducibility, and accountability, and will release full implementation details for independent evaluation.

## Reproducibility Statement

Project code, data, and results are accumulated in the following repository and are shared upon request: `https://github.com/YusukeHashimotoLab/MI_CC`. All datasets, code, and agent logs from the 361 projects are included. The agent framework—roles, decision protocols, and recovery mechanisms—is modular for independent replication. Random seeds are fixed and reported. Experiments include full details (hyperparameters, splits, metrics). The 100-Project Cyclic Theme System and QA protocols are documented for adoption in other domains.

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
